# Origin, structure and functional transition of sex pheromone components in a false widow spider

Andreas Fischer [1✉], Regine Gries[1], Santosh K. Alamsetti[1], Emmanuel Hung[1], Andrea C. Roman Torres[1], Yasasi Fernando [1], Sanam Meraj[1], Weiwu Ren[2,3], Robert Britton[2] & Gerhard Gries[1]

Female web-building spiders disseminate pheromone from their webs that attracts mate-seeking males and deposit contact pheromone on their webs that induces courtship by males upon arrival. The source of contact and mate attractant pheromone components, and the potential ability of females to adjust their web's attractiveness, have remained elusive. Here, we report three new contact pheromone components produced by female false black widow spiders, *Steatoda grossa*: *N*-4-methylvaleroyl-*O*-butyroyl-L-serine, *N*-4-methylvaleroyl-*O*-iso-butyroyl-L-serine and *N*-4-methylvaleroyl-*O*-hexanoyl-L-serine. The compounds originate from the posterior aggregate silk gland, induce courtship by males, and web pH-dependently hydrolyse at the carboxylic-ester bond, giving rise to three corresponding carboxylic acids that attract males. A carboxyl ester hydrolase (CEH) is present on webs and likely mediates the functional transition of contact sex pheromone components to the carboxylic acid mate attractant pheromone components. As CEH activity is pH-dependent, and female spiders can manipulate their silk's pH, they might also actively adjust their webs' attractiveness.

[1] Department of Biological Sciences, Simon Fraser University, Burnaby, BC, Canada. [2] Department of Chemistry, Simon Fraser University, Burnaby, BC, Canada. [3] Present address: School of Medicine and Pharmacy, Ocean University of China, Qingdao, China. ✉email: afischer@sfu.ca

Attracting or finding a mate is essential for all sexually reproductive animal species[1–3]. The process is mediated by long-range communication signals that have chemical, auditory, visual, vibratory or multi-modal characteristics[4–7]. Chemicals such as pheromones are deemed the oldest form of (sexual) communication signals[8] and have evolved in various animal taxa, including mammals[9], myriapods[10], crustaceans[11] and insects[12–15]. Airborne pheromones have signal functions in the context of aggregation[16], territorial marking[17], warning[18], nest defence[19] and reproduction[20,21]. Volatile sex pheromones attract prospective mates[7], whereas cuticle-bound mate recognition pheromones impart reproductive isolation and insect speciation[22].

Sex pheromones have been most extensively studied in insects[12–15]. Beetles, moths, ants and wasps all produce, and release, pheromones from specific glands located in various parts of their body[23]. Many insects can actively time their pheromone production and release, and modulate the amount of pheromone they emit[7,14,24]. Pheromones are perceived by olfactory receptors on the insects' antennae[12] involving complex molecular interactions between pheromone receptors and their pheromone ligands[25–27]. More than 3000 insect pheromones have already been identified[13]. Using the insects' antennae as an analytical tool to help locate candidate pheromone components in complex analytical samples[28] has been instrumental in identifying many of these pheromones, particularly those that occur at trace quantities[29]. In contrast, to date, only 12 spider sex pheromones have been identified and neither their site of production nor their site of reception is known[30,31].

Web-building spiders are multi-modal communicators, using primarily pheromonal and vibratory communication signals[32]. Pheromones play major roles during habitat selection[30,33,34], mate competition[35,36], courtship[37] and mate choice[33,38]. Unlike insects that typically disseminate pheromones from specific gland tissues[7,24], female spiders deposit pheromones on their silken webs[39]. Their webs attract males over long distances[40] and, upon contact, elicit courtship in males[41], implying the release of mate attractant pheromone components from the web and the presence of contact pheromone components on the web[39]. To date, it is not known whether (i) spider pheromones originate from a silk gland, (ii) mate attractant and contact pheromone components are structurally and functionally related, and (iii) female spiders can actively modulate the release of mate attractant pheromone components from their webs.

Pheromone components that female spiders deposit on their webs and that induce courtship by males upon contact have been identified in the linyphiid spider *Linyphia triangularis*[42] and the widow spiders *Latrodectus hasselti*[43] and *L. hesperus*[44]. Female *L. triangularis* deposit (R)-3-[(R)-3-hydroxybutyryloxy]-butyric acid (**1**) on their webs, whereas female *L. hasselti* and *L. hesperus* deposit serine derivatives [*N*-3-methyl-butyryl-*O*-(S)-2-methyl-butyryl-L-serine methyl ester[43] (**2**); *N*-3-methylbutanoyl-*O*-methylpropanoyl-L-serine methyl ester[44] (**3**)(Fig. 1a)]. Both **1** and its breakdown monomer, (R)-3-hydroxybutyric acid (**4**), induce courtship by male *L. triangularis*[31,42]. These results imply that the breakdown of contact pheromone components could engender more volatile pheromone components that then attract males. We predicted that a potential breakdown of *Latrodectus* serine methyl esters could be catalysed by a carboxyl ester hydrolase, which was found on *L. hesperus* webs[45]. As enzyme activity is pH-dependent[46], and spider females might be able to adjust their silk's pH[47], we surmised that *Latrodectus* females possibly modulate the breakdown dynamics of their serine methyl ester deposits, and thus the release of their mate attractant pheromone components.

Here we worked with the globally invasive and synanthropic false black widow spider, *Steatoda grossa* (Theridiidae,

Araneae)[48]. *Steatoda grossa* inhabits predominantly buildings, where it reproduces year-round irrespective of season[48,49]. As *Steatoda* and *Latrodectus* spiders are close phylogenetic relatives[50–52], we anticipated that *S. grossa* would produce pheromone components structurally resembling those of *Latrodectus*. We report the identification of *S. grossa* contact pheromone components, their origin, and breakdown to volatile mate attractant pheromone components, likely catalysed by a pH-dependent carboxyl ester hydrolase present on the females' webs.

## Results and discussion

**Identification of contact pheromone components.** To obtain analyte for the identification of contact pheromone components, we allowed 93 sexually mature adult virgin females and—for comparative analysis—70 sexually immature subadult females[37] three days to build their webs on a prism scaffold (Fig. 1b), building upon previous results that only mature females produce pheromone components[37]. We then methanol-extracted pooled webs from each of the two female groups[37] and analysed extracts by gas chromatography–mass spectrometry (GC-MS). These analyses revealed seven compounds (**5–11** in Fig. 1d; pyrrolidin-2-one (**5**), 4-hydroxyhydrofuran-2(3*H*)-one (**6**), nonanoic acid (**7**), dodecanoic acid (**8**), 6-methylheptanamide (**9**), octanamide (**10**), 4,6-dimethyl heptanamide (**11**), S-Table 1) that were unique to sexually mature females. To test compounds **5–11** for their ability to induce courtship by male spiders, we treated one piece of filter paper on a T-rod apparatus (Fig. 1c) with a synthetic blend of **5–11** (Exp. 1), or with web extract (positive control; Exp. 2), and the corresponding filter paper with solvent control. As only web extract, but not the blend of **5–11**, elicited courtship by males ($N_1 = N_2 = 20$, W = 370, $P < 0.001$, Exp. 1 + 2, Fig. 1e;), it follows that **5–11** are not contact pheromone components. Concerned that the contact pheromone components were too polar or too large to chromatograph in GC-MS analyses, we fractionated web extract by high-performance liquid chromatography (HPLC) and bioassayed each of 20 HPLC fractions for courtship responses by males on the T-rod apparatus. All fractions that elicited courtship behaviour by males (Supplementary Fig. 2) were then analysed by HPLC-tandem mass spectrometry (MS/MS) and by nuclear magnetic resonance ($^1$H NMR) spectroscopy. HLPC-MS/MS analyses revealed an unknown compound (**12**) with a molecular formula of $C_{13}H_{23}NO_5$ and fragmentation ions 186, 274 (M + 1) and 296 (M + Na), indicating a molecular weight of 273 (Fig. 2a). Both the molecular formula and the molecule's weight matched those of the serine methyl ester (**2**) in web extracts of *L. hesperus* (Fig. 1a). Yet, the $^1$H NMR spectrum of unknown **12** (Supplementary Fig. 2) did not support an ester functionality, and GC-MS analyses of *S. grossa* web extracts did not provide any evidence for the presence of a serine methyl ester. Predicting then that **12** was an acid (rather than an ester) which—due to its polar nature—would not chromatograph well in GC–MS analyses, we esterified crude web extract with trimethylsilyldiazomethane[53] and reanalysed aliquots of this extract by GC-MS. These analyses revealed not only one, but three serine methyl ester derivatives (Fig. 2b; *N*-4-methylvaleroyl-*O*-butyroyl-L-serine methyl ester (**13**), *N*-4-methylvaleroyl-*O*-isobutyroyl-L-serine methyl ester (**14**), and *N*-4-methylvaleroyl-*O*-hexanoyl-L-serine methyl ester (**15**)), supporting our prediction that female *S. grossa* produce serine derivatives with a carboxyl (acid) rather than a methyl ester functionality. To infer the structure of the unknown acid **12**, we drew on evidence that its 186 mass fragment (Fig. 2a) was also present in serine methyl esters **2** and **3** produced by *L. hasselti* and *L. hesperus* (Fig. 1a). For the 186 mass fragment of **12**, this meant that the acyl bound to the nitrogen atom had six carbon atoms, instead of five (as in

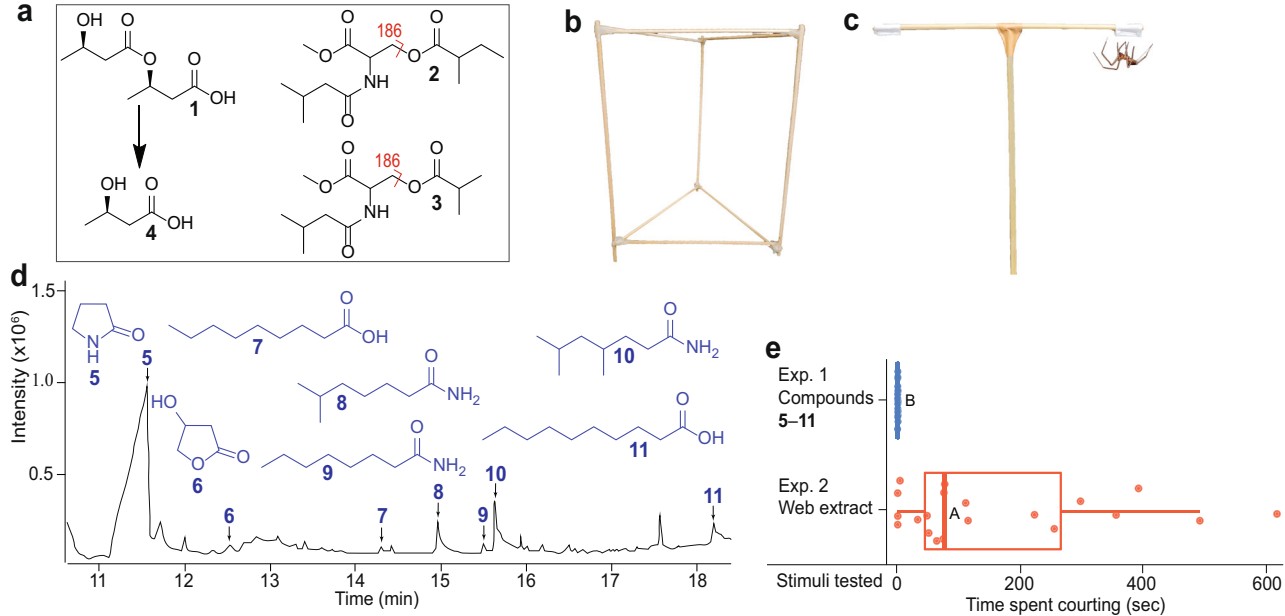

**Fig. 1 Known contact pheromone components of spiders and methods to identify analogous components produced by *Steatoda grossa* females.**
**a** Pheromone components of the spiders (i) *Linyphia triangularis* ([(*R*)-3-hydroxybutyryloxy-butyric acid (**1**) with its breakdown product (*R*)-3-hydroxybutyric acid (**4**)], (ii) *Latrodectus hasselti* ([*N*-3-methyl-butyryl-*O*-(*S*)-2-methylbutyryl-ʟ-serine methyl ester (**2**)], and (iii) *Latrodectus hesperus* [(*N*-3-methylbutanoyl-*O*-methylpropanoyl-ʟ-serine methyl ester (**3**)]. **b** Triangular prism scaffold for a female spider to build her web. **c** T-rod apparatus for testing courtship behaviour by *S. grossa* males in response to test stimuli (web extract or fractions thereof; synthetic candidate pheromone components; solvent control) applied to a piece of filter paper attached to each distal end of the horizontal arm. **d** Total ion chromatogram of compounds unique to sexually mature *S. grossa* females (pyrrolidin-2-one (**5**), 4-hydroxyhydrofuran-2(3*H*)-one (**6**), nonanoic acid (**7**), dodecanoic acid (**8**), 6-methylheptanamide (**9**), octanamide (**10**), 4,6-dimethyl heptanamide (**11**); Supplementary Table 1) identified by gas chromatography–mass spectrometry of crude female web extract. **e** Extent of courtship by *S. grossa* males in response to female web extract or synthetic candidate pheromone components. Circles and boxplots show the time single male spiders courted in each replicate and the distribution of data (minimum, first quartile, median, third quartile, maximum), respectively. Medians with different letters indicate statistically significant differences in courtship responses. Wilcoxon test, $P < 0.05$.

esters **1** and **2**), with 4 possible isomers: 2-, 3- or 4-methylpentanoyl and hexanoyl. For the molecular ion of **12** to be $m/z$ 173, the second acyl bound to the oxygen atom had to have only four carbon atoms with either butyryl or isobutyryl configuration. Of eight possible synthetic isomers (see SI), only *N*-4-methylpentanoyl-*O*-butyryl-ʟ-serine (**12**, Fig. 2a) had HPLC-MS/MS spectrometric and retention characteristics entirely consistent with *S. grossa* produced **12**. Moreover, the corresponding synthetic methyl ester of **12**, *N*-4-methylpentanoyl-*O*-butyryl-ʟ-serine methyl ester, had retention and mass spectral characteristics entirely consistent with those of the most abundant serine methyl ester **13** in esterified web extracts of *S. grossa* (Fig. 2b).

All three serine methyl ester derivatives had similar mass spectra (Fig. 2c), indicating a conserved molecular structure with differences only in the acyl groups of the molecules. Ester **13** [retention index (RI): 1843] and ester **14** (RI: 1890) had identical mass spectra (Fig. 2c), but their RI differential of 43 units indicated a methyl branch in **13**. The RI of ester **15** (2074) was about 200 RI units higher than that of ester **14**, implying the presence of a higher homologue with two additional carbon atoms. To assign definitive molecular structures to esters **13** and **15**, we synthesised multiple standards (see Supplementary material Methods: Syntheses). Of these, *N*-4-methylvaleroyl-*O*-isobutyroyl-ʟ-serine methyl ester and *N*-4-methylvaleroyl-*O*-hexanoyl-ʟ-serine methyl ester had mass spectrometric and retention characteristics entirely consistent with those of the serine methyl ester derivatives **13** and **15**, respectively, in esterified web extracts. Moreover, in HPLC-MS/MS analyses, the corresponding synthetic acids (*N*-4-methylvaleroyl-*O*-isobutyroyl-ʟ-serine (**12**); *N*-4-methylvaleroyl-*O*-isobutyroyl-ʟ-serine

(**16**); *N*-4-methylvaleroyl-*O*-hexanoyl-ʟ-serine (**17**); Fig. 2a) had retention times and mass spectra entirely consistent with those produced by female *S. grossa* and present in web extract. In T-rod (Fig. 1c) bioassays, a ternary blend of the synthetic acids **12**, **16** and **17**, tested at one web equivalent, elicited courtship behaviour by *S. grossa* males comparable to web extract (Exp. 3 vs. Exp. 4: $N_1 = N_2 = 20$, $Z = -0.39$, $P = 0.521$, Fig. 2d), indicating that all essential contact pheromone components were present in this synthetic blend. The seven volatile components **5–11** unique to sexually mature females (Fig. 1d) did not enhance the behavioural activity of the ternary acid blend (**12**, **16** and **17**) (Exp 4. *vs*. Exp. 5: $N_1 = N_2 = 20$, $Z = 0.03$, $P = 0.488$, Fig. 2d) nor did they induce any courtship behaviour on their own (Exp. 6, Fig. 2d). In contrast, the ternary acid blend induced courtship behaviour in a dose-dependent manner (Exps. 7–11: $\chi^2 = 61.75$, df = 4, $P < 0.001$; Supplementary Fig. 3). Binary blends of the acids also induced courtship behaviour, but their effect differed according to blend constituents (Exps. 12–15: $\chi^2 = 11.19$, df = 3, $P = 0.010$; Supplementary Fig. 4). Acids **12** and **16** tested singly elicited courtship as effectively as in binary combination (Exps. 16–18: $\chi^2 = 3.65$, df = 2, $P = 0.160$; Supplementary Fig. 5).

**Origin of contact pheromone components.** Silk glands have been hypothesised[54], but never been experimentally shown, to produce sex pheromones. Moreover, the specific silk gland (out of eight possible glands) that produces the pheromone components has never been determined. With the *S. grossa* contact pheromone components now identified and key spectrometric data of the most abundant component (**12**) in hand (Fig. 2a), we proceeded to trace its origin. For all analyses, we cold-euthanized

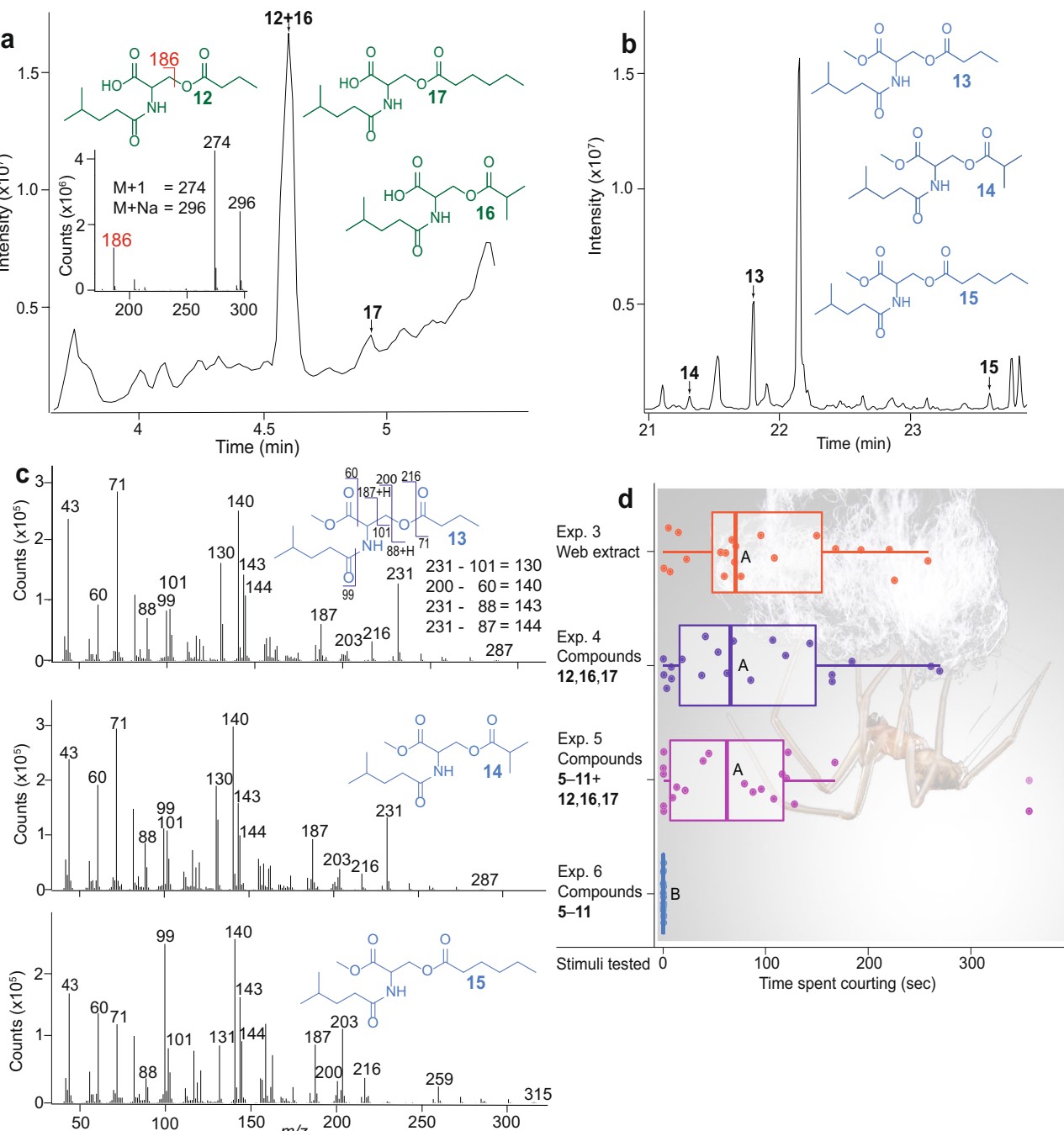

**Fig. 2 Contact pheromone components of female *Steatoda grossa*. a** High-performance liquid chromatogram (HPLC) of compounds [*N*-4-methylvaleroyl-*O*-isobutyroyl-L-serine (**12**); *N*-4-methylvaleroyl-*O*-isobutyroyl-L-serine (**16**); *N*-4-methylvaleroyl-*O*-hexanoyl-L-serine (**17**)] present in crude web extract of female *S. grossa*, and HPLC mass spectrum of **12** (with **16** coeluting). **b, c** Total ion chromatogram (**b**) and mass spectra (**c**) of compounds [*N*-4-methylvaleroyl-*O*-butyroyl-L-serine methyl ester (**13**), *N*-4-methylvaleroyl-*O*-isobutyroyl-L-serine methyl ester (**14**), *N*-4-methylvaleroyl-*O*-hexanoyl-L-serine methyl ester (**15**)] identified by gas chromatography–mass spectrometry (GC-MS) in esterified web extract of female *S. grossa*. **d** Extent of courtship by male *S. grossa* in response to stimuli tested in T-rod bioassays. The names of compounds **5–11** are reported in the caption of Fig. 1. Circles and boxplots show the time single male spiders courted in each replicate and the distribution of data (minimum, first quartile, median, third quartile, maximum), respectively. Medians with different letters indicate statistically significant differences in courtship responses; Kruskal–Wallis $\chi^2$ test with Benjamini–Hochberg correction to account for multiple comparisons, $P < 0.05$.

spiders, extracted body tissue in a methanol/saline solution[55], centrifuged extracts, and analysed aliquots of each tagma or tissue sample by HPLC-MS for the presence and quantity of **12** and **16**. Because contact pheromone components **12** and **16** coeluted in these analyses, we quantified their combined amount. As only the abdomen, but not the cephalothorax, of spiders contained **12** and

**16** (Exp. 19: $N = 22$, $W = 21$, $P = 0.004$, Fig. 3a), we then screened abdominal haemolymph and five specific abdominal tissues, including all eight silk glands combined, for the presence of **12** and **16**. With only silk gland samples containing **12** and **16** (Exp. 20: $N = 20$, $\chi^2 = 70.96$, df = 6, $P < 0.001$, Fig. 3b), we analysed glands separately and found that it was the posterior

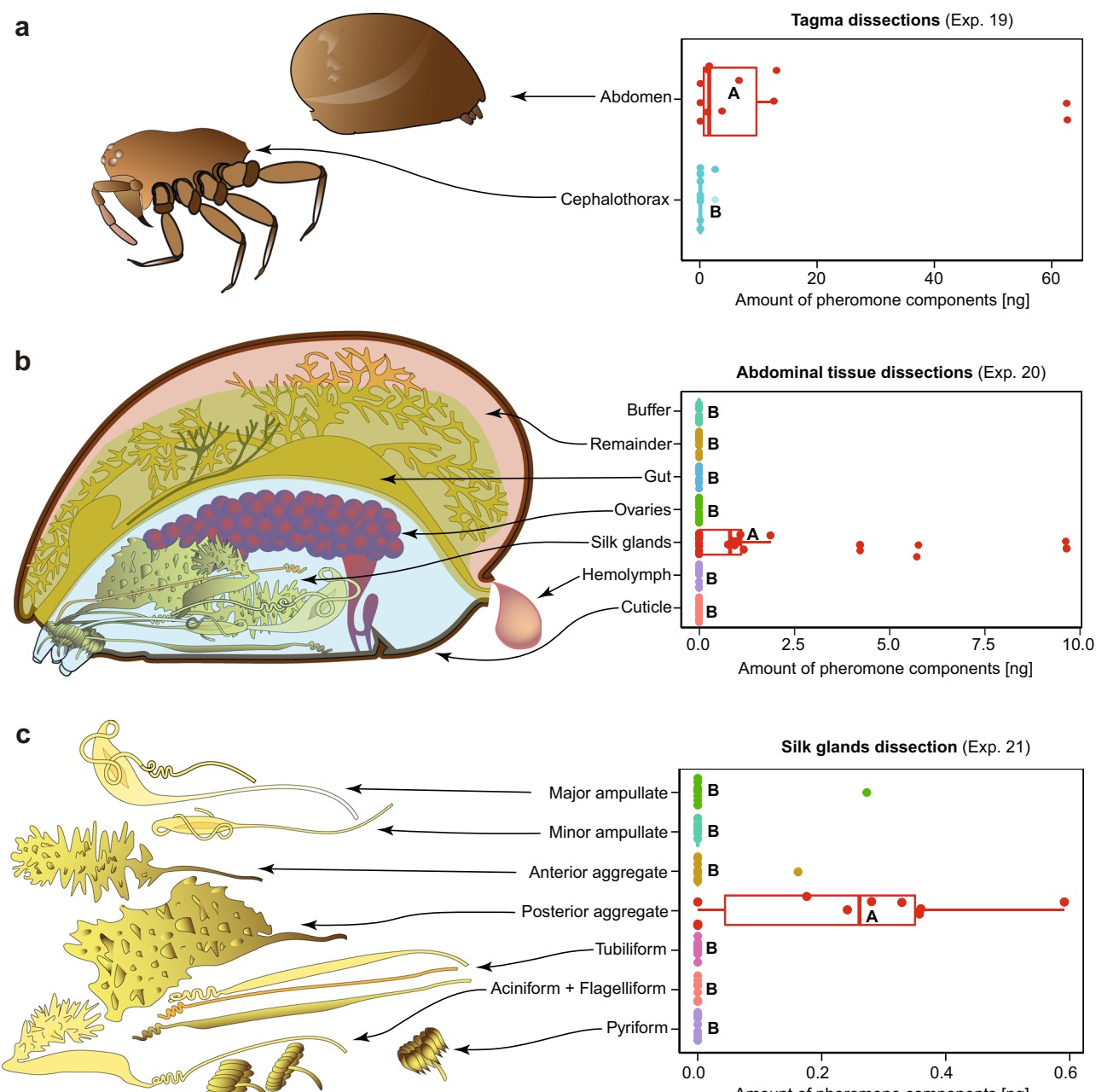

**Fig. 3 Origin of contact pheromone components produced by female *Steatoda grossa*. a** High-performance liquid chromatography–mass spectrometry (HPLC-MS) quantification of two contact pheromone components [*N*-4-methylvaleroyl-*O*-isobutyroyl-ʟ-serine (**12**) coeluting with *N*-4-methylvaleroyl-*O*-isobutyroyl-ʟ-serine (**16**)], present in the abdomen and cephalothorax of female spiders. **b** HPLC-MS quantification of **12** and **16** in the haemolymph and various tissues of the abdomen. **c** HPLC-MS quantification of **12** and **16** in various silk glands. In each of experiments 19–21, circles and boxplots show the amount of **12** and **16** present in each spider and the distribution of data (minimum, first quartile, median, third quartile, maximum), respectively. Medians with different letters indicate significantly different amounts of **12** and **16** present in various sources; Wilcoxon and Kruskal–Wallis $\chi^2$ test with Benjamini–Hochberg correction to account for multiple comparisons, $P < 0.05$.

aggregate gland that exclusively, or most abundantly, contained **12** and **16** (Exp. 21: $N = 30$, $\chi^2 = 36.00$, df = 6, $P < 0.001$, Fig. 3c). Although not specifically tested, it is likely that the posterior aggregate gland also produces contact pheromone component **17**.

**Transition of contact pheromone components to volatile mate attractant pheromone components.** Long-distance orientation of male spiders to mate attractant pheromone components emanating from female *S. grossa* webs was tested in Y-tube, moving-air olfactometers[56], using web extract (instead of webs) as the test

stimulus (Fig. 4a). When offered a choice between web extract and a solvent control, males were attracted to web extract (Exp. 22: $N = 21$, $P = 0.013$, Fig. 4b). However, when offered a choice between the blend of volatile compounds **5–11** unique to sexually mature females (Fig. 1d) and a solvent control, males exhibited no attraction responses (Exp. 23: $N = 20$ $P = 0.588$, Fig. 4b). These data implied that the mate attractant pheromone components were not readily detectable and possibly arose from chemical reactions occurring on the web. Drawing on a previous report[31,42] that the dimer contact pheromone **1** of the spider *L. triangularis* breaks down to a volatile monomer attractant (**4**) (Fig. 1a), we

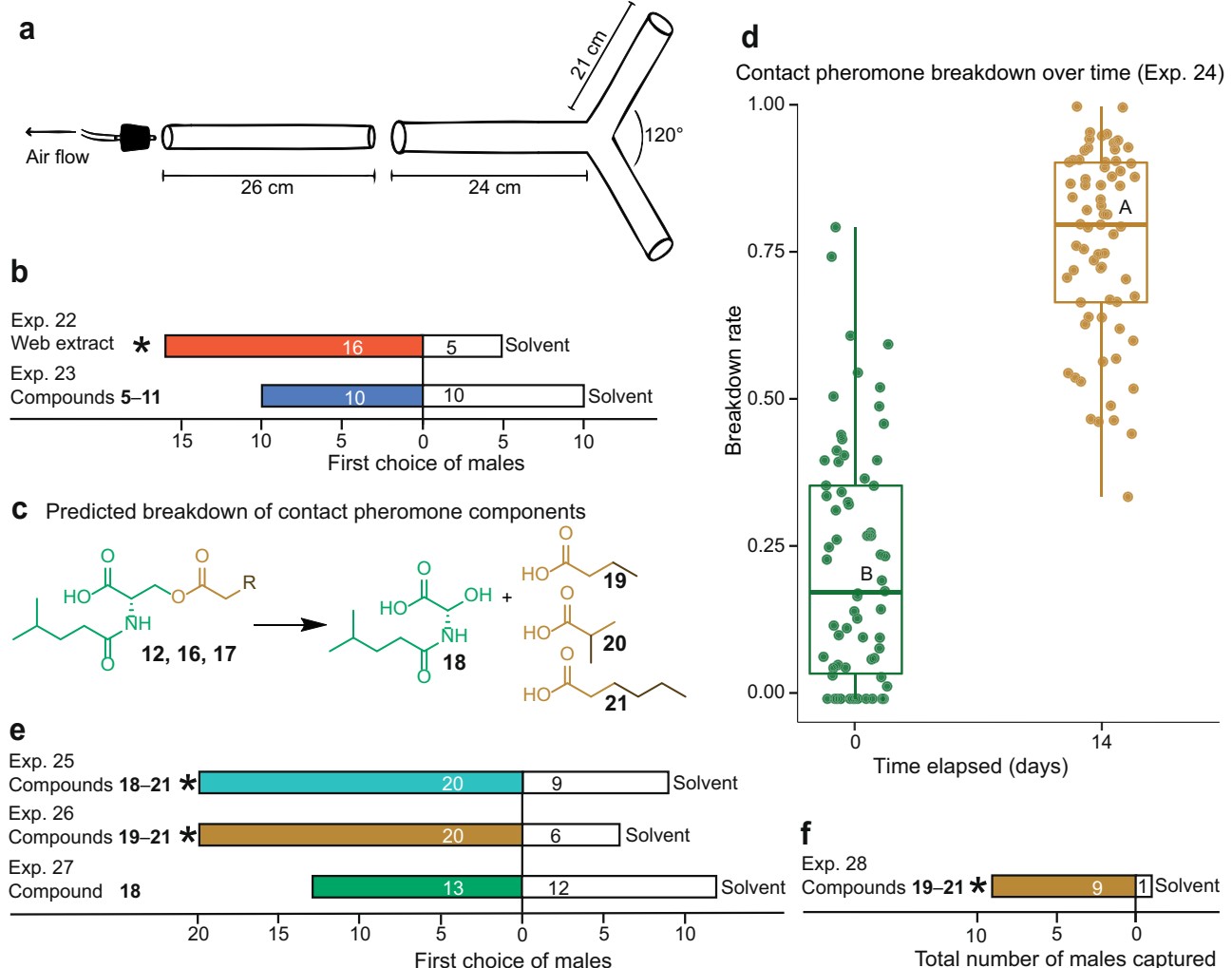

**Fig. 4 Transition of contact pheromone components produced by female *Steatoda grossa* to sex attractant pheromone components. a** Moving-air dual-choice Y-tube olfactometer. **b** Attraction of *S. grossa* males in Y-tube olfactometers to extracts of female webs and to volatile compound **5–11** (names in Fig. 1 caption) unique to sexually mature females. **c** Predicted breakdown of contact pheromone components **12**, **16**, and **17** to the amide **18** and the volatile carboxylic acid mate attractant pheromone components **19**, **20**, and **21**. **d** Breakdown rate of contact pheromone components [ratio of **18**/ (**12** + **16** + **17** + **18**)] on webs extracted 0 or 14 days after being built; circles and boxplots show the breakdown rates of single webs and the distribution of data (minimum, first quartile, median, third quartile, maximum), respectively, at days 0 and 14, which differed significantly (Wilcoxon test, $P < 0.05$). **e** Attraction of *S. grossa* males in Y-tube olfactometers to single- or multiple-component blends of synthetic compounds; in each experiment, an asterisk denotes a significant preference for the treatment stimulus (one-tailed binomial tests; $P < 0.05$). **f** Captures of *S. grossa* males in ten pairs of sticky traps that were deployed in building hallways between September and December 2018. During weekly checks, the position of the treatment and control trap within each pair was randomised; the treatment trap was baited with the carboxylic acids **19**, **20**, and **21** (see Methods for detail), whereas the control trap was left unbaited; the asterisk denotes a significant preference for the treatment trap (one-tailed binomial test; $P < 0.05$).

hypothesised (Fig. 4c) that the contact pheromone components **12**, **16** and **17** of female *S. grossa* hydrolyse over time at the carboxylic-ester bond, giving rise to the amide *N*-4-methylvaler-oyl-L-serine (**18**) and three corresponding carboxylic acids [butyric (**19**), isobutyric (**20**), hexanoic (**21**)], and that these volatile acids then attract males. Realising the difficulty to quantify the release of these acids over time, we instead quantified the accumulating amide **18** as a proxy for the breakdown of contact pheromone components (Fig. 4c). Our breakdown hypothesis was supported by data showing a significantly higher breakdown ratio [**18**/(**18** + **12** + **16** + **17**)] in extracts of 14-day-old webs than in those of freshly spun (0-day-old) webs (Exp. 24: $W = 637$, $N_{0 \text{ days}} = N_{14 \text{ days}} = 70$, $p < 0.001$, Fig. 4d). Moreover, our attraction hypothesis was supported by Y-tube olfactometer data (Fig. 4e) showing that males are attracted to a blend of the four breakdown products **18–21** (Exp. 25: $N = 29$, $P = 0.030$) and

to a blend of the three carboxylic acids **19–21** (Exp. 26: $N = 26$, $P = 0.006$), but not to the amide **18** (Exp. 27: $N = 25$, $P = 0.500$). Tested on its own, amide **18** also did not elicit any courtship behaviour by males, nor did it increase the activity of the contact pheromone components **12**, **16** and **17**, which—when tested as a ternary blend in parallel—effectively induced courtship (Exp. 29–31: $\chi^2 = 12.78$, df = 2, $P < 0.001$, Supplementary Fig. 6)

To substantiate our conclusion that the carboxylic acids **19–21** function as mate attractant pheromone components of female *S. grossa*, we formulated these acids in mineral oil[9] and tested them as a trap lure in building hallways with low *S. grossa* infestations. Over the course of 16 weeks, carboxylic acid-baited traps captured nine *S. grossa* males, whereas corresponding control traps captured only one male, confirming the mate attractant pheromone function of the carboxylic acids (Exp. 28: $N = 10$, $P = 0.011$, Fig. 4f).

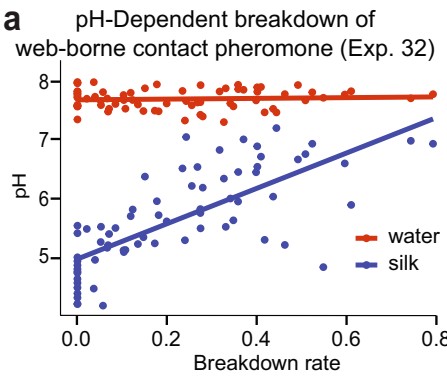

**a** pH-Dependent breakdown of
web-borne contact pheromone (Exp. 32)

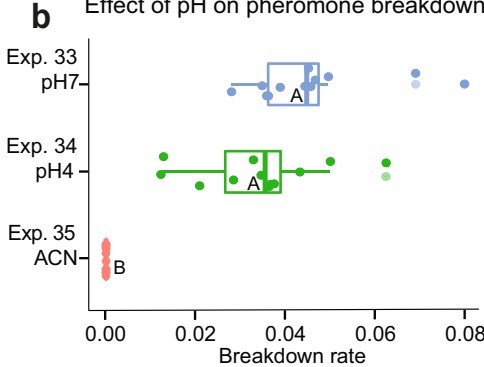

**b** Effect of pH on pheromone breakdown

**Fig. 5 pH-Dependent breakdown of contact pheromone components.**
**a** Relationship between the pH of female *Steatoda grossa* webs and the
breakdown rate of contact pheromone components **12**, **16**, and **17**,
calculated as the ratio of **18**/(**12** + **16** + **17** + **18**) (in blue); control
measurements of the water's pH are displayed in red. **b** Effect of pH on the
breakdown of synthetic contact pheromone component **12**, calculated as
the ratio of **18**/(**12** + **18**). Circles and boxplots show the breakdown rate of
each sample and the distribution of data (minimum, first quartile, median,
third quartile, maximum), respectively, at pH 4 and pH 7; medians with the
same letter indicate no significant difference in breakdown rates
(Kruskal–Wallis test; $P < 0.05$). Note the different scales of the x-axis in
subpanels **a** and **b**; **12** = *N*-4-methylvaleroyl-*O*-isobutyroyl-L-serine; **18** = *N*-
4-methylvaleroyl-L-serine (amide), ACN acetonitrile.

**Mechanisms underlying the transition of contact pheromone
components to mate attractant pheromone components.** We
hypothesised that the chemical breakdown of the contact pher-
omone components **12**, **16** and **17**, and the release of the car-
boxylic acids **19**, **20** and **21** as mate attractant pheromone
components are catalysed by one or both of two non-mutually
exclusive mechanisms: (1) the activity of a web-borne carboxyl
ester hydrolase (CEH) and (2) direct saponification of the contact
pheromone components. Both mechanisms are pH-dependent.
To test our prediction that breakdown rates of contact pher-
omone components are positively correlated with the webs' pH,
we allowed each of 70 spiders to spin two webs. We used one web
from each spider to quantify the contact pheromone components
(**12**, **16** and **17**) and their amide breakdown product (**18**), and the
other web to determine its pH. For pH measurements, we
determined the pH of each web by adding the web to a small
volume of water which served as a conductor for the pH metre[57].
Plotting the data revealed a significant positive correlation
between the pH of webs and the chemical breakdown rate (ratio
of **18**/(**12** + **16** + **17** + **18**)) (Exp. 32: $F_{1,69} = 108.44$, $p < 0.001$;
Fig. 5a).

To further determine whether pH directly affects the hydrolysis
of contact pheromone components, we exposed synthetic **12** to

pH 7 or pH 4 buffer solutions, and to aprotic acetonitrile. While
both buffer solutions afforded significantly greater breakdown
rates than the aprotic control solution (Exp. 33–35, $\chi^2 = 25.84$,
df = 2, $p < 0.001$, Fig. 5b), the effect size was 10-fold lower than
that measured on webs (Exp. 24). Thus, the pH as a direct (single)
factor is insufficient to catalyse the hydrolysis of contact
pheromone components (**12**, **16**, **17**) to mate attractant
pheromone components (**19–21**). However, if there were a
carboxyl ester hydrolase (CEH) to be present on *S. grossa* webs, as
there is on *L. hesperus* webs[45], then pH could affect the enzymatic
activity of a CEH, and thereby the hydrolysis of contact
pheromone components. With *S. grossa* and *L. hesperus* being
close phylogenetic relatives[51], we predicted that they produce not
only similar serine-derived contact pheromone components[39]
(see **3** and **12**) but also a similar or identical CEH to hydrolyse
them. To test our prediction, we extracted webs in Sørensen
buffer[58] from three groups of spiders: (1) adult virgin female *L.
hesperus* (positive control, known to have a CEH[45]); (2) subadult
sexually immature female *S. grossa* (deemed to have not yet
produced a CEH) and (3) adult virgin sexually mature female *S.
grossa* (predicted to have the same CEH as *L. hesperus*). To
account for different amounts of silk produced by these three
groups of spiders, we extracted five webs of *L. hesperus*, 20 webs
of subadult *S. grossa* and ten webs of adult *S. grossa* in each of
three replicates, and submitted extracts for comparative proteo-
mics (CEH analyses) (see SI for detailed methods). The CEH was
present in all three samples of *L. hesperus* and adult *S. grossa* and
—surprisingly—also in two of three samples of subadult *S. grossa*,
possibly because some webs were produced by females about to
become sexually mature. Conceivably, this CEH may—pH—
dependently—hydrolyse the *S. grossa* contact pheromone com-
ponents, with females manipulating enzyme activity by altering
their webs' pH. Increasing their webs' pH would enhance the
hydrolysis of contact pheromone components (Fig. 5b), and, thus,
the release of mate attractant pheromone components, making
their webs more attractive to mate-seeking males. This concept
could be tested experimentally. Once engineered CEH becomes
available, it could be placed on artificial (Halloween) spider web[30]
with specific pH values and treated with synthetic contact
pheromone component to measure hydrolysis rates.

Our study addresses significant questions about the commu-
nication ecology of web-building spiders. These unresolved
questions were whether (1) spider pheromone originates from a
silk gland, (2) mate attraction and courtship-inducing contact
pheromone components are chemically interlinked and (3)
female spiders actively adjust pheromone dissemination from
their web to attract males. Here, we provide definitive answers to
questions 1 and 2, and we discuss data for question 3. First, we
identified three previously unknown serine-derived contact
pheromone components produced by *S. grossa* females: *N*-4-
methylvaleroyl-*O*-isobutyroyl-L-serine (**12**); *N*-4-methylvaleroyl-
*O*-isobutyroyl-L-serine (**16**); *N*-4-methylvaleroyl-*O*-hexanoyl-L-
serine (**17**). We then show that these components originate from
the posterior aggregate silk gland and—once web-borne—induce
courtship by males. We further demonstrate a functional
transition of these contact sex pheromone components to volatile
mate attractant pheromone components. Web pH-dependently,
the contact pheromone components hydrolyse at the carboxylic-
ester bond and give rise to three corresponding carboxylic acids
that attract males. With increasing web pH (4–7), hydrolysis rates
increase and greater amounts of carboxylic acids (as hydrolysis
products) are released. However, pH 7 alone is insufficient to
induce biologically significant hydrolysis rates. Subjecting syn-
thetic contact pheromone to a pH 7 buffer solution induced
hydrolysis rates tenfold lower than those measured on webs.
These data imply that the hydrolysis is catalysed by an enzyme,

most likely the carboxyl ester hydrolase that is present on *S. grossa* webs. This carboxyl ester hydrolase, pH-dependently, might hydrolyse the contact pheromone components, with the enzyme apparently being most active around pH 7. Our explanation of enzyme-catalysed contact pheromone hydrolysis is supported, in part, by pheromone studies of the widow spider *L. hesperus*, a phylogenetically close relative of *S. grossa*[50]. Female *L. hesperus* also produce a serine derivative contact pheromone component[44] (Fig. 1a) that is likely hydrolysed by a carboxyl ester hydrolase, reported to be present on *L. hesperus* webs[45].

Sustained dissemination of mate attractant pheromone components from a reservoir of web-borne contact pheromone components is adaptive for sessile web-building spiders. Sustained pheromone dissemination establishes a somewhat permanent information flow to potential signal recipients. This type of dissemination system is reminiscent of pheromone dissemination from urine markings of murine rodents. Here, major urinary proteins bind to mate attractant pheromone components, and facilitate their slow release[59], thus prolonging the attractiveness of pheromonal markings[60].

If we accept the concept that in *S. grossa* an enzyme is involved in mediating the transition of contact pheromone components to mate attractant pheromone components, and if we apply the common knowledge that enzyme activity is pH-dependent[46], and spiders lower the pH in their spinning apparatus to convert aqueous silk to solid silk threads[47,61,62], it follows that female *S. grossa* might be able to actively adjust their web's attractiveness to males. To date, only insects were known to actively time their pheromone production and dissemination[63], and to modulate the amount of pheromone they emit[64]. Our findings suggest, but do not prove, that web-building spiders might do this as well. With the pheromone system of *S. grossa* now known, potential manipulation by female spiders of their webs' pH, and thus their webs' attractiveness to mate-seeking males, can now be tested in the context of honest or dishonest signalling.

Our finding that the posterior aggregate silk gland is the source of contact pheromone components in *S. grossa* will help expedite pheromone identification in other spiders, provided—of course—that their pheromones originate from the same silk gland. Pheromone-producing glands often contain a sufficiently large amount of pheromone analyte for structural elucidation[7]. Many insect pheromones could be identified primarily because the pheromone-producing gland was known, and many glands could be extracted for pheromone accumulation and analysis[65,66]. For pheromone identification in web-building spiders, it would also be easier to extract and analyse the content of the pheromone-producing silk gland than to extract and analyse an entire web with, possibly, many more constituents.

## Conclusions

In conclusion, our study reveals the intricate pheromonal communication system of *S. grossa*, as a model species for web-building spiders, and it provides an incentive for comparative studies in other spider taxa.

## Methods

**Experimental spiders**. Experimental spiders were maintained as previously reported[37]. Briefly, spiders were the F1 to F4 offspring of mated females collected from hallways of the Burnaby campus of Simon Fraser University (Burnaby, BC, CA). Upon hatching, juvenile spiders were housed individually in petri dishes (100 mm × 20 mm) and provisioned with the vinegar flies *Drosophila melanogaster*. Subadult spiders were fed with larvae of the mealworm beetle *Tenebrio molitor*. Each adult female spider was kept in a separate translucent 300-mL plastic cup (Western Family, CA) maintained at 22 °C under a reversed light cycle (12:12 h). Adult males and females were fed with black blow flies, *Phormia regina*. All spiders had access to water in cotton wicks. Water and food were provided once per week. Laboratory experiments were run during a reversed scotophase (0900 to 1700).

**Identification of contact pheromone components: Preparation of web extracts (summer 2017; spring and summer 2018)**. Each of the 100 spiders was allowed to build her web for three days on a wooden triangular prism scaffold (30 cm × 25 cm × 22 cm)[44] of bamboo skewers (GoodCook, CA, USA) (Fig. 1b). After the spiders were removed from the scaffold, their webs were reeled up with a glass rod (10 cm × 0.5 cm) and deposited in a 1.5-mL glass vial. Per web, 50 µL of methanol (99.9% HPLC grade, Fisher Chemical, ON, Canada) were added and the silk was extracted for 24 h at room temperature. Prior to analysis, the silk was removed and the sample was concentrated under a steady nitrogen stream to the desired concentration.

**Identification of contact pheromone components: analyses of web extracts by gas chromatography–mass spectrometry (GC-MS)**. Aliquots (2 µL) of pooled and concentrated web extract (100 webs in 400 µL of solvent) were analysed by GC–MS, using a Varian Saturn Ion trap 2000 (Varian Inc., now Agilent Technologies Inc., Santa Clara, CA 95051, USA) and an Agilent 7890B GC coupled to a 5977 A MSD, both fitted with a DB-5 GC-MS column (30 m × 0.25 mm ID, film thickness 0.25 µm). The injector port was set to 250 °C, the MS source to 230 °C, and the MS quadrupole to 150 °C. Helium was used as a carrier gas at a flow rate of 35 cm s$^{-1}$, with the following temperature programme: 50 °C held for 5 min, 10 °C min$^{-1}$ to 280 °C (held for 10 min). Compounds were identified by comparing their mass spectra and retention indices (relative to aliphatic alkanes[67]) with those of authentic standards that were purchased or synthesised in our laboratory (Supplementary Table 1).

**Identification of contact pheromone components: high-performance liquid chromatography (HPLC) of web extracts**. Web extract of virgin adult female *S. grossa* was fractionated by high-performance liquid chromatography (HPLC), using a Waters HPLC system (Waters Corporation, Milford, MA, USA; 600 Controller, 2487 Dual Absorbance Detector, Delta 600 pump) fitted with a Synergy Hydro Reverse Phase $C_{18}$ column (250 mm × 4.6 mm, 4 µ; Phenomenex, Torrance, CA, USA). The column was eluted with a 1-mL/min flow of a solvent gradient, starting with 80% water (HPLC grade, EMD Millipore Corp., Burlington, MA, USA) and 20% acetonitrile (99.9% HPLC grade, Fisher Chemical, Ottawa, CA) and ending with acetonitrile after 10 min. A 60-web-equivalent extract was injected and 20 1-min fractions were collected. Each HPLC fraction (containing 20 web-equivalents) was tested in T-rod bioassays (Fig. 1c) for the presence of contact pheromone components. All eight fractions that elicited courtship responses by males (Supplementary Fig. 1) were analysed by HPLC-tandem MS/MS.

**Identification of contact pheromone components: HPLC-tandem MS/MS of bioactive HPLC fractions**. The bioactive HPLC fractions were analysed on a Bruker maXis Impact Quadrupole Time-of-Flight HPLC/MS System. The system consists of an Agilent 1200 HPLC fitted with a spursil $C_{18}$ column (30 mm × 3.0 mm, 3 µ; Dikma Technologies, Foothill Ranch, CA, USA) and a Bruker maXis Impact Ultra-High Resolution tandem TOF (UHR-Qq-TOF) mass spectrometer. The LC-MS conditions were as follows: The mass spectrometer was set to positive electrospray ionisation (+ESI) with a gas temperature of 200 °C and a gas flow of 9 L/min. The nebuliser was set to 4 bar and the capillary voltage to 4200 V. The column was eluted with a 0.4-mL/min flow of a solvent gradient, starting with 80% water and 20% acetonitrile and ending with 100% acetonitrile after 4 min. The solvent contained 0.1% formic acid to improve peak shape.

**Identification of contact pheromone components: $^{1}$H NMR analyses of a bioactive fraction**. In HPLC-MS analyses, a single bioactive fraction (9–10 min) appeared to contain only a single compound. This fraction was then further investigated using $^{1}$H NMR spectroscopy. The $^{1}$H NMR spectrum was recorded on a Bruker Advance 600 equipped with a QNP (600 MHz) using CDCl$_3$. Signal positions (δ) are given in parts per million from tetramethylsilane (δ 0) and were measured relative to the signal of the solvent ($^{1}$H NMR: CDCl$_3$: δ 7.26).

**Identification of contact pheromone components: syntheses of candidate pheromone components**. The syntheses of candidate pheromone components and synthetic intermediates are reported in the SI.

**Identification of contact pheromone components: T-rod bioassays (general procedures)**. The T-rod apparatus[37] (Fig. 1c) consisted of a horizontal beam (25 cm × 0.4 cm) and a vertical beam (30 cm × 0.4 cm) held together by labelling tape (3 cm × 1.9 cm, Fisher Scientific, Ottawa, ON, CA). A piece of filter paper (2 cm$^2$) was attached to each distal end of the horizontal beam. For each bioassay, an aliquot of web extract (in methanol), or a blend of synthetic candidate pheromone components, was applied to the randomly assigned treatment filter paper, whereas methanol was applied to the control filter paper. The solvent was allowed to evaporate for 1 min before the onset of a 15-min bioassay. A randomly selected naïve male spider was placed at the base of the vertical beam and the time he spent courting on each filter paper was recorded. In response to the presence of female-produced or synthetic pheromone on a filter paper, the male engaged in courtship, pulling silk with his hindlegs from his spinnerets and adding it to the paper. Sensing contact pheromone, the male essentially behaves as if he were courting on

the web of a female. On a web, the male engages in web reduction prior to copulation, a behaviour that entails cutting sections of the female's web with his chelicerae and wrapping the dismantled web bundle with his own silk pulled from his spinnerets[41,56]. Each T-rod apparatus was used only once. Replicates of experiments as part of specific research objectives were run in parallel to eliminate day effects on the responses of spiders. The sample size for each experiment was set to 20 unless otherwise stated.

**Identification of contact pheromone components: T-rod bioassays (specific experiments) (fall 2017; spring and summer 2018).** Experiment 1 (fall 2017) tested a synthetic blend of volatile compounds **5–11** unique to mature *S. grossa* females (Fig. 1c and Supplementary Table 1) *vs* a solvent control. Parallel experiment 2 tested one web equivalent of virgin female web extract, followed by testing each of the 20 HPLC fractions in six replicates for the occurrence of courtship (spring 2018).

Parallel experiments 3–6 (summer 2018) tested web extract at one female web equivalent (1 FWE) (Exp. 3), a ternary blend of the candidate contact pheromone components **12**, **16** and **17** (Fig. 2d, Exp. 4), the same ternary blend (**12**, **16** and **17**) in combination with the volatile compounds **5–11** (Exp. 5), and **5–11** on their own (Exp. 6).

Parallel dose-response experiments 7–11 (summer 2018) tested the ternary blend of **12**, **16** and **17** at five FWEs: 0.001 (Exp. 7); 0.01 (Exp. 8); 0.1 (Exp. 9); 1.0 (Exp. 10); and 10 (Exp. 11).

Parallel experiments 12–15 tested the ternary blend, and all possible binary blends, of **12**, **16** and **17**. Parallel experiments 16–18 tested **12** and **16** in binary combination (Exp. 16) and singly (Exps. 17, 18).

**Origin of contact pheromone components (fall 2020).** To trace the origin of contact pheromone component **12** (and coeluting **16**), cold-euthanized female spiders were dissected in saline solution[55] (25 mL of water and 25 mL of methanol, 160 mM NaCl, 7.5 mM KCl, 1 mM MgCl₂, 4 mM NaHCO₃, 4 mM CaCl₂, 20 mM glucose, pH 7.4). Samples were homogenised (Kimble Pellet Pestle Motor, Kimble Kontes, USA) in methanol for 1 min, kept 24 h at room temperature for pheromone extraction, and then centrifuged (12,500 rpm, 3 °C for 20 min; Hermle Z 360 K refrigerated centrifuge; B. Hermle AG, Wehingen, DE) to obtain the supernatant for HPLC-MS analyses (see above) for the presence of **12** and **16**. Three sequential sets of dissections aimed to determine (1) the pheromone-containing body tagma, (2) the pheromone-containing tissues or glands in that tagma and (3) the specific gland or tissue producing **12** & **16**.

To identify the pheromone-containing tagma, 11 spiders were severed at the pedicel, generating two tagmata: the cephalothorax with four pairs of legs and the abdomen. Each tagma was then extracted separately in 100 µL of methanol. Eight of 11 abdomen samples contained **12** and **16**, whereas only one of 11 thorax samples contained **12** and **16** (Exp. 19), albeit at only trace amounts. With **12** and **16** being present in the abdomen, 20 additional abdomens were dissected[68] to obtain separate samples of (i) haemolymph (25 µL), (ii) ventral cuticle (~0.5 cm² near the pedicel, (iii) the ovaries, (iv) all silk glands combined, and (v) the gut (with anus, cloaca and Malpighian tubules). The remaining spider tissues (vi) were pooled as one sample, and 20 µL of the dissection buffer solution (vii) was obtained to detect potential pheromone bleeding. To each tissue sample, 50 µL of methanol were added. Only silk gland samples contained **12** and **16** (Exp. 20). Having established that only silk gland samples contained **12** and **16**, the silk glands of 30 additional spiders were excised in the following order: (i) major ampullate gland, (ii) minor ampullate gland, (iii) anterior aggregate gland, (iv) posterior aggregate gland, (v) tubuliform, (vi) aciniform and flagelliform glands combined and (vii) pyriform gland. The glands from three spiders were combined in each sample and extracted in 30 µL methanol. Seven of ten posterior aggregate gland samples contained **12** and **16**, with other silk gland samples not containing **12** and **16** or in only trace amounts (Exp. 21).

**Transition of contact pheromone components to mate attractant pheromone components: evidence for hydrolysis of contact pheromone components (12, 16 and 17) (spring 2021).** To test for the hydrolysis of the contact pheromone components **12**, **16** and **17**, we compared their breakdown ratio (**18**/(**12** + **16** + **17** + **18**) on independent webs aged 0 days and 14 days (Exp. 24). Each of 140 spiders was allowed to spin a web on bamboo scaffolds for three days. Then, the spiders were removed and webs—by random assignment—were extracted immediately (0-day-old webs) or after 14 days of aging (14-day-old webs). On each web, the amount of contact pheromone components **12**, **16** and **17**, and of amide **18** as a breakdown product, was quantified using HPLC–MS, with **12** and **18** at 25 and 50 ng/µL as external standards.

**Transition of contact pheromone components to mate attractant pheromone components: Y-tube olfactometer bioassays (general procedures).** The attraction of male spiders to web extracts and to candidate mate attractant pheromone components was tested in Y-tube olfactometers[56] (Fig. 4a) lined with bamboo sticks to provide grip for the bioassay spider. Test stimuli were presented in translucent oven bags (30 cm × 31 cm; Toppits, Mengen, DE) secured to the orifice of side-arms. Test stimuli consisted of a triangular bamboo prism scaffold (each side 8.5 cm long)

bearing a spider's web, or bearing artificial webbing[30] (40 ± 2 mg; Bling Star, CN) that was treated with web extract or synthetic chemicals in methanol (100 µL) as the treatment stimulus or with methanol (100 µL) as the control stimulus. For each experimental replicate, a male spider was introduced into a glass holding tube and allowed 2 min to acclimatise. Then, the holding tube was attached via a glass joint to the Y-tube olfactometer and an air pump was connected to the holding tube, drawing air at 100 mL/min through the olfactometer. Air entered the olfactometer through a glass tube secured to the oven bags' second opening. A male that entered the olfactometer within the 5-min bioassay period was classed a responder and his first choice of oven bag (the oven bag he reached first) was recorded. Whenever a set of 30 replicates was completed by the same observer, using 30 separate Y-tubes, the Y-tubes were cleaned with hot water and soap (Sparkleen, Thermo Fisher Scientific, MA, United States) and dried in an oven at 100 °C for 3 h, whereas the bamboo sticks and the oven bags were discarded.

**Transition of contact pheromone components to mate attractant pheromone components: Y-tube olfactometer bioassays (specific experiments) (summer 2018).** In experiments 22, 23 and 25–27, males were offered a choice between a solvent control stimulus and a treatment stimulus. The treatment stimulus consisted of (i) virgin female web-extract (1 web-equivalent) (Exp. 22, *N* = 24), (ii) the volatile compounds **5–11** unique to sexually mature females (Fig. 1d) (Exp. 23, *N* = 24), (iii) all breakdown products of the contact pheromone components **12**, **16** and **17**, consisting of the amide *N*-4-methylvaleroyl-L-serine (**18**) and the corresponding carboxylic acids **19**, **20** and **21** (Exp. 25, *N* = 30), (iv) a blend of the acids **19**, **20** and **21** (Exp. 26, *N* = 30) and (v) the amide **18** as a single compound (Exp. 27, *N* = 30). Compounds were tested at quantities as determined in virgin female web extract (50 webs in 150 µL of dichloromethane), following silyl-ester derivatization[69] of acids in the extract, with valeric acid (200 ng; ≥99%, Sigma Aldrich, St. Louis, USA) added as an internal standard. Per web equivalent, there were 103 ng of **19**, 3 ng of **20** and 54 ng of **21**. The amide **18** was present at 200 ng per web equivalent, as determined using *N*-3-methylbutnaoyl-L-serine methyl ester as an external standard.

**Transition of contact pheromone components to mate attractant pheromone components: hallway of buildings experiment (fall 2018).** As the ternary blend of the carboxylic acids **19**, **20** and **21** attracted male spiders in Y-tube olfactometers (see Results), we aimed to confirm their functional role as mate attractant pheromone components also in 'field' settings (Exp. 28). To this end, we set up ten replicates of paired traps in building hallways on the Burnaby campus of Simon Fraser University. Adhesive-coated traps (Bell Laboratories Inc., Madison, WI, USA) were spaced 0.5 m within pairs and 20 m between pairs. By random assignment, one trap in each pair was baited with the carboxylic acids **19**, **20** and **21** formulated in 200 µL of mineral oil (Anachemia, Montreal, CA; 2.8 mg of **19**, 0.112 mg of **20** and 1.52 mg of **21**), whereas the control trap received mineral oil only. Test stimuli were disseminated from a 400-µL microcentrifuge tube (Evergreen Scientific, Ranco Dominguez, CA, USA) with a hole in its lid punctured by a No. 3 insect pin (Hamilton Bell, Montvale, NJ, USA). Every week for 4 months (September to December 2018), traps were checked, lures were replaced, and the position of the treatment and the control trap within each trap pair was re-randomised.

**Communication function of amide breakdown product 18 (fall 2018).** As the amide **18** did not attract males in Y-tube olfactometer experiments (see Results), we tested its alternate potential function as a contact pheromone component which, if active, would induce courtship by males. Using the T-rod apparatus (Fig. 1c), we treated one piece of filter paper with a solvent control and the other with a blend comprising both the contact pheromone components **12**, **16** and **17** and the amide **18** (Exp. 29), a blend of **12**, **16** and **17** (Exp. 30), and **18** alone (Exp. 31).

**Mechanisms underlying the transition of contact pheromone components to mate attractant pheromone components: relationship between web pH and breakdown rates of contact pheromone components (summer 2020).** We allowed each of the 70 spiders to spin two webs, using one web to quantify the amide breakdown product (**18**) of the contact pheromone components (see above), and the other web to determine its pH according to the slurry method[57] (Exp. 32). To this end, we first measured the pH of 50 µL water (HPLC grade, EMD Millipore Corp., Burlington, MA, USA) and then of a web with the water functioning as a conductor for the pH metre (LAQUAtwin pH 22 (Horiba, Kyoto, JP). Between web measurements, the pH metre was rinsed with water and regularly re-calibrated using a pH 7 and a pH 4 buffer (Horiba, Kyoto, JP).

**Mechanisms underlying the transition of contact pheromone components to mate attractant pheromone components: testing for pH-dependent saponification of contact pheromone components (12, 16 and 17) (summer 2021).** To test whether pH alone catalyses saponification of the ester bond of contact pheromone components (**12**, **16** and **17**), synthetic **12** was added to a 40% aqueous pH 7 buffer solution (Exp. 34), a pH 4 buffer solution (Exp. 34), and to acetonitrile (Exp. 35) as a polar aprotic solvent control (*N* = 12; 100 ng/µL each). pH-Dependent breakdown of **12** over time was assessed by analysing (HPLC-MS)

diluted aqueous aliquots (2.5 ng/µl) of each sample at day 0 and after 14 days of storage at room temperature.

**Mechanisms underlying the transition of contact pheromone components to mate attractant pheromone components: testing for the presence of a carboxylesterhydrolase (CEH) (summer 2021).** To test for the presence of a carboxylesterhydrolase (CEH), for each of three replicates we extracted (i) five webs of adult virgin female *L. hesperus* (positive control, known to have a CEH[45]), (ii) 20 webs of subadult *S. grossa* (deemed to have not yet produced a CEH) and (iii) ten webs of adult virgin female *S. grossa*, accounting for the different amounts of silk produced by these three groups of spiders. For each replicate, webs were extracted in 200 µL 0.05 M Sørensen buffer[58] and analysed by Bioinformatics Solutions (Waterloo, ON, CA). After web samples were incubated for 20 min at 60 °C in 2× sample volumes of 10% SDS (lauryl sulfate; protein-denaturing anionic detergent), they were sonicated for 20 min. Then, the supernatant was withdrawn, reduced with dithiothreitol (DTT), and alkylated with iodoacetamide (IAA). Alkylated samples were treated further with a *protein* solvent (S-Trap kit; Protifi, Farmingdale, NY, USA). Briefly, samples were acidified by phosphoric acid to pH ≤1. Then 6× of sample volume S-trap buffer was mixed in. The mixture was loaded by centrifugation onto an S-Trap Micro Spin Column and washed 3× with S-trap buffer. Using the serine protease trypsin, protein digestions were carried out at 47 °C for 1 h in 50 mM triethylamonium bicarbonate (TEAB) buffer within the S-Trap Micro Spin column. Digestion products were eluted sequentially with 40 µL 50 mM TEAB and 0.2% formic acid. Eluates were dried and re-suspended in 0.1% formic acid.

Eluates were analysed by HPLC-MS/MS in positive ion mode on a Thermo Scientific Orbitrap Fusion Lumos Tribrid mass spectrometer (Thermo Fisher, San Jose, CA, USA), equipped with a nanospray ionisation source and a Thermo Fisher Ultimate 3000 RSLCnano HPLC System (Thermo Fisher). Peptide mixtures were loaded onto a PEPMAP100 C$_{18}$ trap column (75 µm × 20 mm, 5 µm particle size; Thermo Fisher) at a constant flow of 30 µL/min and 60 °C isothermal. Peptides were eluted at a rate of 0.2 µL/min and separated using a Reprosil C$_{18}$ analytical column (75 µm × 15 mm, 1.9 µm particle size; PepSep, DK) with a 60-min solvent gradient: 0–45 min: 4–35% acetonitrile + 0.1% formic acid; 45–55 min: 90% acetonitrile + 0.1% formic acid; 55–60 min: 4% acetonitrile + 0.1% formic acid.

MS data were acquired in data-dependent mode with a cycle time of 3 s. MS1 scan data were acquired with the Orbitrap mass analyser, using a mass range of 400–1600 *m/z*, with the resolution set to 120,000. The automatic gain control (AGC) was set to 4e5, with a maximum ion injection time of 50 ms, and the radio frequency (RF) lens was set to 30%. Isolations for MS2 scans were run using a quadrupole mass analyser, with an isolation window of 0.7. MS2 scan data were acquired with the Orbitrap mass analyser at a resolution of 15,000 *m/z*, with a maximum ion injection time of 22 ms, and the AGC target set to 5e4. Higher energy collisional dissociation (HCD; fixed normalised collision energy: 30%) was used for generating MS2 spectra, with the number of microscans set to 1.

**Statistics and reproducibility.** Data (Supplementary Table 2) were analysed statistically using R[70]. Data of experiments 1–18 and 29–31 (testing courtship by male spiders in response to contact pheromone components) were analysed with a Wilcoxon test or Kruskal–Wallis two-tailed rank-sum test with Benjamini–Hochberg correction to adjust for multiple comparison. Data of experiments 19–21 (revealing the presence of contact pheromone components in the abdomen, silk glands, and posterior aggregate silk gland) were analysed with two-tailed, rather than one-tailed, Wilcoxon test or Kruskal–Wallis rank tests because we had no strong assumption as to whether or not pheromone would be present in any of these potential pheromone sources. The *p* values were adjusted for multiple comparison using the Benjamini–Hochberg method. Y-tube olfactometer data of experiments 22, 23 and 25–27, as well as the hallway experiment 28 (revealing attraction of male spiders to volatile pheromone components) were analysed using an one-tailed[71] binomial test, anticipating attraction of spiders to volatile mate attractant pheromone components rather than to solvent control stimuli. Data of experiment 32 (revealing a correlation between web pH and breakdown of web-borne contact pheromone components) were analysed using generalised linear models. Data of experiments 33–35 (showing pH-dependent breakdown of synthetic contact pheromone) were compared using a two-tailed Kruskal–Wallis test with Benjamini–Hochberg correction.

**Reporting summary.** Further information on research design is available in the Nature Research Reporting Summary linked to this article.

## Data availability
All data, including raw data underlying the figures, can be found in Supplementary Data 1.

## Code availability
All codes that were used to analyse the data can be found in Supplementary Data 1.

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

## Acknowledgements

We thank three anonymous reviewers for constructive comments, Hongwen Chen, Adam Blake and Catherine McCaughey for technical advice, and Stephen Takács for preparing Fig. 3. Further, we thank our funders: A.F.: Graduate Fellowship, McCarthy Bursary from Simon Fraser University, Alexander Graham Bell Scholarship from the Natural Sciences and Engineering Research Council of Canada (NSERC); G.G.: NSERC—Industrial Research Chair with Scotts Canada Ltd. and BASF Canada as the industrial sponsors. The funders had no role in study design, data collection and analysis, decision to publish or preparation of the manuscript.

## Author contributions

Conceptualisation: A.F., G.G. and R.G. Data curation, formal analysis, software, validation and visualisation: A.F. Funding acquisition: G.G. Investigation: A.F., R.G., S.K.A., E.H., A.C.R.T., Y.F., S.M., W.R., R.B. and G.G. Methodology and writing—original draft preparation: A.F. and G.G. Project administration: A.F., R.G. and G.G. Resources: G.G. and R.G. Supervision: G.G. and A.F. Writing— review and editing: G.G., A.F. and E.H.

## Competing interests

The authors declare no competing interests.
