## [Peer Review File · Communications Biology]

Reviewers' comments:

Reviewer #1 (Remarks to the Author):

This is a beautiful piece of work, novel and interesting for a wide public. With an enormous dataset and an interdisciplinary approach, the authors have provided good evidence that pheromones in this species originate from a silk gland, and that mate attraction and courtship-inducing pheromones components are chemically interlinked. Less supported but also relevant, the authors also suggest that female spiders actively adjust pheromone dissemination from their web to attract males. Personally, as an ethologist working on arachnids, I am very happy to finally get to know where spider pheromones are produced. We have had lots of behavioral papers on the subject, identifications of the chemical components, but the site of production has always been a missing piece. I thank the authors for that!

My main concerns are a few but relevant ones. Methods of the behavioral parts are poorly described, making it hard to know what was done. Depending on how the experiments were conducted, there may be serious issues with the data. Or maybe the authors simply forgot to mention a number of important information. First, I apologize in advance if what I found missing was actually mentioned, the manuscript + supp material has more than 50 pages. Years, months, hours of the day, alternation of treatments, cleaning of arenas between trials, operational variables for "choice", none of these were mentioned. Such information is standard in behavioral journals because they are very relevant, especially for seasonal, nocturnal animals that leave a dragline behind when they walk. One can easily have wrong results if these were overlooked. My second main comment is that the conclusion that this spider can actively adjust their web's attractiveness to males seems to be less supported than the other two main conclusions of the manuscript. Using the authors own words, "If we accept the concept that in *S. grossa* an enzyme is involved in mediating the transition of contact pheromone components to mate attractant pheromone components, and apply the common knowledge that enzyme activity is pH dependent⁴² and spiders manipulate their webs' pH⁴³, it follows that female *S. grossa* can actively adjust their web's attractiveness to males." Therefore, I would suggest tone it down a little bit.

Other comments and detailing of some of the comments above are find below.

We know that hour of the day influences behavioral reactions. Nocturnal animals may not do anything during the day. The same goes for month of the year, animals not in their reproductive period may not respond to conspecific chemicals. The number of experiments in this study is astonishing, sometimes with high sample sizes. I wonder how much time was needed to run all the trials and I suspect several months, probably a couple of years were necessary. If control animals were tested during the day and treatment animals at night, for example, obviously there will be a difference. I did not find information of hour, month or year neither in Methods nor in Supplementary Material. This comment applies for all behavioral experiments.

L2 – false widows are not in the same genera as widows. The title is misleading as is and should read "pheromone components in false widow spider"

L54 "It is also not known how mate-seeking males sense pheromones³⁷." The authors are describing web building spiders in the paragraph. Still, for the sake of precision, I would specify that this phrase refers to web building spiders, because known contact chemoreceptors on the pedipalps of males are known to sense female pheromones on the dragline of cursorial spiders: TICHY, H., GINGL, E., EHN, R., PAPKE, M. & SCHULZ, S. (2001). Female sex pheromone of a wandering spider (*Cupiennius salei*): identification and sensory reception. *Journal of Comparative Physiology A* 187, 75–78.

L64–74 I found it odd that predictions are made to *Latrodectus* and the manuscript is about *Steatoda*, with the justification that *Steatoda* "... is closely related to *Latrodectus* widow spiders". The authors could explain that better to avoid surprising the reader.

Scientists around the world have been putting effort into valorizing the importance of women in science. We now worry about inviting women as keynote speakers, provide help for pregnant scientists etc. Citations are also relevant. French scientist Marie Trabalon was one of the first scientists (if not the first) to study the relevance of HCs in spider communication, extending her studies to the importance of chemicals on the web in intraspecific recognition. The authors could consider citing one of her studies. MARIE TRABALON, ANNE GENEVIEVE. Contact sex signals in two sympatric spider species, *Tegenaria domestica* and *Tegenaria pagana*. *Journal of Chemical*

Ecology, 23, 1997: 747-758; M. TRABALON & D. ASSI-BESSEKON Effects of web chemical signatures on intraspecific recognition in a subsocial spider, *Coelotes terrestris* (Araneae). *Animal Behaviour*, 2008, 76, 1571-1578.

L110 The authors have tested a syntethic blend of "seven compounds (5–11 in Fig. 1d; pyrrolidin-2-one (5), 4-hydroxyhydrofuran-2(3H)-one (6), nonanoic acid (7), dodecanoic acid (8), 6-methylheptanamide (9), octanamide (10), 4,6-dimethyl heptanamide (11),". What was the proportion of each chemical in the blend?

L130 "supporting our prediction that female *S. grossa* produce serine derivatives with a carboxyl rather than a methyl ester functionality". But a few lines above, in 124, it is written that "Predicting then that 12 was an acid (rather than an ester) which – due to its polar nature – would not chromatograph well in GC-MS analyses, we esterified crude web extract with trimethylsilyldiazomethane⁴⁸ and reanalysed aliquots of this extract by GC-MS". Reading "supporting our prediction", the reader is led to believe the authors are referring to the last prediction just made, but they are not. Maybe rephrase.

318-322 – The third item (adjusting pheromone dissemination) relies on a number of assumptions as the authors themselves recognize (L 349-352). Therefore, I think this could be rephrased.

351- "and spiders manipulate their webs' pH⁴³" Because this is a very important part of the manuscript and the rest of the paragraph relies on this being true, the authors should elaborate on it.

354 – "Our findings indicate that web-building spiders can do this too" Please tone it down a bit. These spiders might be able to do it.

355-369 I found this paragraph somewhat speculative, not needed in such a high level and well supported manuscript

If the chemicals that trigger courtship transition the chemicals that attract mates, does it mean that older webs will attract mates but will not trigger courtship? How long does a web of *Steatoda grossa* last in nature? Please address the issue.

461 - "the male engaged in courtship, pulling silk with his hindlegs from his spinnerets and adding it to the paper." This sentence deserves a citation and probably a very brief explanation. Pulling silk as a proxy of courtship will probably sound weird, since only those working on the taxon and subject know the sexual behavior of theridiids and others of silk wrapping. Please cite the works of Catherine Scott and/or Andreas Fischer on courtship in *Steatoda grossa*.

514-521 – It seems that the authors have used the same spiders in day 1 and day 14. "After female spiders had spun their webs on bamboo scaffolds for three days, they were removed and webs were extracted immediately (0-day-old webs) or following 14 days of aging (14-day-old webs)." If that is the case, please specify that you have used Wilcoxon signed rank (matched pair design) and not Wilcoxon rank sum (independent samples).

537 – Spiders leave a dragline, often with pheromones, when they walk. In the Y olfactometer experiment, the authors did not mention cleaning the apparatus after each trial. They also did not mention how they scored the trials. What is the operational variable for "being attracted" (L 234)? Each researcher uses a distinct criterium. Often, arthropods go to the end of one of the arms of the Y maze, go back, move to the end of the other arm and remains there, for example.

637 – The authors have used, in a single case, a one tailed test. They have justified it with the citation of a reference. Please elaborate on it. Other tests ran in the study had a "one-tailed" context, expectation, a hypothesis that generate a prediction of one group having smaller or larger values than the other. But in such cases, two tailed tests were used.

Reviewer #2 (Remarks to the Author):

Fischer et al. identified the chemicals of contact sex pheromones of a widow spider species and demonstrated their roles in attracting males. This study also provides evidence showing that the breakdown of a contact pheromone leads to chemicals with higher volatility which can attract males over a distance. This is a very cool study and the manuscript is well written with clear visualization. I only have some minor comments for this manuscript and I recommend a minor revision.

Minor comments.

- Technically speaking, spiders do not belong to insects. While in the second paragraph of the Introduction, the authors mainly introduce pheromone-related literature in insects but ignored

findings in a broader range of species in arthropods. I suggest the authors have a more inclusive introduction on arthropods, instead of only focusing on insects.

- Line 53-54, this study only showed that pH could lead to different efficiencies in the breakdown of contact pheromone but not how female spiders actively modulate this breakdown. I suggest rephrasing this sentence to make it clearer.
- Line 55, this study did not investigate how mate-seeking males sense pheromones. I suggest the authors delete this sentence.
- Figure 5, how about putting panel b to the left side of panel a?

Reviewer #3 (Remarks to the Author):

This is a very nice paper on the pheromone communication of the false black widow spider. The authors identified a non-volatile sex pheromone produced by females and applied onto the web to induce courtship in males. The exiting and new part of the study is, that the same non-volatile compounds are catalyzed by an enzyme into a volatile form to attract males over longer distances. Thus, the paper adds not only a lot of new information on the pheromone communication of spiders but reveals a new mechanism that connect the two pheromones and produces the volatile form. Also, from an evolutionary point of view, this is very interesting. Technically, the paper is well done. The authors put a lot of work in the chemical analysis and especially synthesis of the pheromone compounds. Also, the behavioral experiments are quite extensive.

I only have a few minor comments, mostly on figures and statistical analysis:

Fig. 1e: If the statistic is ranked based, i.e. a Kruskal-Wallis or Wilcoxon test, I think it is not really appropriate to show mean and SE, but better to give a box plot (combined with datapoints). This also applies Fig. 2d, 4d and 5b.

Fig. 2d: You obviously also use pairwise Wilcoxon Test after the Kruskal-Wallis test. Did you also use a correction for multiple comparisons? Also, the background picture is nice, but does not add any information. Please remove.

Line 196 and Fig. 3b. The 140 samples have been taken from 20 spiders and are thus not independent. Better write $N=20$.

Line 198: Same as above. $N=30$.

Line 210 and Fig. 3: Did you use Fisher's exact test for a), b) and c)? Why this test and not a Kruskal-Wallis? If you used Fisher's exact test, you did only test the presence or absence of the compounds, right? This should be stated. And does not really fit with a figure showing the amounts and means.

Fig. 4f: You put up 10 traps (line 555), right? So $N=10$, not 160. Furthermore, what is shown in the figure? The mean per trap or the total number of spiders caught in all traps? Which statistics did you use?

Line 261: Move the reference for table S1 right behind the compound numbers.

Responses (R) to reviewers' Comments

R1: Please note: all references to line numbers refer to the manuscript with tracked changes

Reviewers' comments:

Reviewer #1 (Remarks to the Author):

2. This is a beautiful piece of work, novel and interesting for a wide public. With an enormous dataset and an interdisciplinary approach, the authors have provided good evidence that pheromones in this species originate from a silk gland, and that mate attraction and courtship-inducing pheromones components are chemically interlinked. Less supported but also relevant, the authors also suggest that female spiders actively adjust pheromone dissemination from their web to attract males. Personally, as an ethologist working on arachnids, I am very happy to finally get to know where spider pheromones are produced. We have had lots of behavioral papers on the subject, identifications of the chemical components, but the site of production has always been a missing piece. I thank the authors for that!

R2: We thank Reviewer #1 for the kind remarks.

3. My main concerns are a few but relevant ones. Methods of the behavioral parts are poorly described, making it hard to know what was done. Depending on how the experiments were conducted, there may be serious issues with the data. Or maybe the authors simply forgot to mention a number of important information. First, I apologize in advance if what I found missing was actually mentioned, the manuscript + supp material has more than 50 pages. Years, months, hours of the day, alternation of treatments, cleaning of arenas between trials, operational variables for “choice”, none of these were mentioned. Such information is standard in behavioral journals because they are very relevant, especially for seasonal, nocturnal animals that leave a dragline behind when they walk. One can easily have wrong results if these were overlooked.

R3: Thank you for having pointed out missing information. We are confident that all experiments were carefully designed and that there was no cross-contamination between replicates. We agree, though, that some important information was not conveyed. Wherever indicated, we have added this information to the revised manuscript. In the Experimental Spider section, we have added (L421f): “Laboratory experiments were run during a reversed scotophase (0900 to 1700)”. It is noteworthy that *S. grossa* inhabits buildings and does not experience seasonal changes. Over the many years we have been working on *S. grossa*, we have found its

reproductive behaviour not to be correlated with seasons. We have added an exploratory statement to the introduction (L74ff): “*Steatoda grossa* inhabits predominantly buildings, where it reproduces year-round irrespective of season”. Even though time of year had no effect on reproductive activities of *S. grossa*, we have added information to the revised manuscript as to when pheromone samples were prepared and behavioural bioassays were run and added it to the respective headline. Furthermore, we better describe the response criterion in Y-tube olfactometer bioassays and how we have avoided cross-contamination between replicates, as follows (569ff): “A male that entered the olfactometer within the 5-min bioassay period was classed a responder and his first choice of oven bag (the oven bag he reached first) was recorded. Whenever a set of replicates was completed, the Y-tubes were cleaned with hot water and soap (Sparkleen, Thermo Fisher Scientific, MA, United States) and dried in an oven at 100 °C for three hours, whereas the bamboo sticks and the oven bags were discarded”.

4. My second main comment is that the conclusion that this spider can actively adjust their web’s attractiveness to males seems to be less supported than the other two main conclusions of the manuscript. Using the authors own words, “If we accept the concept that in *S. grossa* an enzyme is involved in mediating the transition of contact pheromone components to mate attractant pheromone components, and apply the common knowledge that enzyme activity is pH dependent⁴² and spiders manipulate their webs’ pH⁴³, it follows that female *S. grossa* can actively adjust their web’s attractiveness to males.” Therefore, I would suggest tone it down a little bit.

R4: We agree that this is particular conclusion is less supported than other conclusions and for that reason had used very tentative/cautionary wording. Nonetheless, in response to this comment, we have further toned down our statement. This section now reads as follows (L374ff): “If we accept the concept that in *S. grossa* an enzyme is involved in mediating the transition of contact pheromone components to mate attractant pheromone components, and if we apply the common knowledge that enzyme activity is pH dependent⁴⁵ and spiders lower the pH in their spinning apparatus to convert aqueous silk to solid silk threads^{47,61,62}, it follows that female *S. grossa* might also be able to actively adjust their webs’ attractiveness to males. To date, only insects were known to actively time their pheromone production and dissemination⁶⁴, and to modulate the amount of pheromone they emit. Our findings suggest, but do not prove, that web-building spiders might do this as well. With the pheromone system of *S. grossa* now known, potential manipulation by female spiders of their webs’ pH, and thus their webs’ attractiveness to mate-seeking males, can now be tested in the context of honest or dishonest signalling”.

5. Other comments and detailing of some of the comments above are find below. We know that hour of the day influences behavioral reactions. Nocturnal animals may not do anything during the day. The same goes for month of the year, animals not in their reproductive period may not respond to conspecific chemicals. The number of

experiments in this study is astonishing, sometimes with high sample sizes. I wonder how much time was needed to run all the trials and I suspect several months, probably a couple of years were necessary. If control animals were tested during the day and treatment animals at night, for example, obviously there will be a difference. I did not find information of hour, month or year neither in Methods nor in Supplementary Material. This comment applies for all behavioral experiments.

R5: Please see R3. All behavioural experiments were run, and all treatment and control stimuli were tested, during the scotophase. Moreover, replicates of many experiments were run in parallel, thus eliminating a day effect on the responses of spiders. Yes, to test our hypotheses we needed to maintain very large spider colonies, run many experiments, and produce large data sets. All these are reasons why the study took multiple years to complete.

6. L2 – false widows are not in the same genera as widows. The title is misleading as is and should read “pheromone components in false widow spider”

R6: We have revised the title accordingly (also condensing it not to exceed the 15-word maximum).

7. L54 “It is also not known how mate-seeking males sense pheromones³⁷.” The authors are describing web building spiders in the paragraph. Still, for the sake of precision, I would specify that this phrase refers to web building spiders, because known contact chemoreceptors on the pedipalps of males are known to sense female pheromones on the dragline of cursorial spiders: TICHY, H., GINGL, E., EHN, R., PAPKE, M. & SCHULZ, S. (2001). Female sex pheromone of a wandering spider (*Cupiennius salei*): identification and sensory reception. *Journal of Comparative Physiology A* 187, 75–78.

R7: This sentence seemed to imply that we also studied olfaction in males (which we did not). Not to mislead the reader, we have deleted this sentence (as also suggested by Reviewer #2).

8. L64-74 I found it odd that predictions are made to *Latrodectus* and the manuscript is about *Steatoda*, with the justification that *Steatoda*... is closely related to *Latrodectus* widow spiders”. The authors could explain that better to avoid surprising the reader.

R8: As suggested, we have added information (L75f): “As *Steatoda* and *Latrodectus* spiders are close phylogenetic relatives^{50–52}, we anticipated that *S. grossa* would produce pheromone components structurally resembling those of *Latrodectus*”.

9. Scientists around the world have been putting effort into valorizing the importance of women in science. We now worry about inviting women as keynote speakers, provide

help for pregnant scientists etc. Citations are also relevant. French scientist Marie Trabalon was one of the first scientists (if not the first) to study the relevance of HCs in spider communication, extending her studies to the importance of chemicals on the web in intraspecific recognition. The authors could consider citing one of her studies. MARIE TRABALON, ANNE GENEVIEVE. Contact sex signals in two sympatric spider species, *Tegenaria domestica* and *Tegenaria pagana*. *Journal of Chemical Ecology*, 23, 1997: 747-758; M. TRABALON & D. ASSI-BESSEKON Effects of web chemical signatures on intraspecific recognition in a subsocial spider, *Coelotes terrestris* (Araneae). *Animal Behaviour*, 2008, 76, 1571-1578.

R9. Thank you for the suggestion. In the revised manuscript, we are citing both of Dr. Trabalon's publications (L49f).

10. L110 The authors have tested a synthetic blend of "seven compounds (5–11 in Fig. 1d; pyrrolidin-2-one (5), 4-hydroxyhydrofuran-2(3H)-one (6), nonanoic acid (7), dodecanoic acid (8), 6-methylheptanamide (9), octanamide (10), 4,6-dimethyl heptanamide (11),". What was the proportion of each chemical in the blend?

R10: The proportion of chemicals had been reported in S-Table 1 which was listed after the string of chemical names. No revision is warranted in response to this comment.

11. L130 "supporting our prediction that female *S. grossa* produce serine derivatives with a carboxyl rather than a methyl ester functionality". But a few lines above, in 124, it is written that "Predicting then that 12 was an acid (rather than an ester) which – due to its polar nature – would not chromatograph well in GC-MS analyses, we esterified crude web extract with trimethylsilyldiazomethane⁴⁸ and reanalysed aliquots of this extract by GC-MS". Reading "supporting our prediction", the reader is led to believe the authors are referring to the last prediction just made, but they are not. Maybe rephrase.

R11: Thank you for having pointed out that the chemical nomenclature may be confusing to readers. To avoid further confusion, we have inserted '(acid)' after 'carboxyl' to indicate that 'carboxyl' and 'acid' are indeed interchangeable terms that refer to the same functionality group.

12. 318-322 – The third item (adjusting pheromone dissemination) relies on a number of assumptions as the authors themselves recognize (L 349-352). Therefore, I think this could be rephrased.

R12: We have rephrased this section as follows (L341ff): "Our study addresses significant questions about the communication ecology of web-building spiders. These unresolved questions were whether (1) spider pheromone originates from a silk gland, (2) mate attraction and courtship-inducing contact pheromone components are chemically interlinked, and (3) female

spiders actively adjust pheromone dissemination from their web to attract males. Here, we provide definitive answers to questions 1 and 2, and we discuss data for question 3.

13. 351- “and spiders manipulate their webs’ pH43” Because this is a very important part of the manuscript and the rest of the paragraph relies on this being true, the authors should elaborate on it.

R13: We have revised the respective sentence and added information, as follows (L374ff): “If we accept the concept that in *S. grossa* an enzyme is involved in mediating the transition of contact pheromone components to mate attractant pheromone components, and if we apply the common knowledge that enzyme activity is pH dependent⁴⁵, and spiders lower the pH in their spinning apparatus to convert aqueous silk to solid silk threads^{47,61,62}, it follows that female *S. grossa* might be able to actively adjust their webs’ attractiveness to males”.

14. 354 – “Our findings indicate that web-building spiders can do this too” Please tone it down a bit. These spiders might be able to do it.

R14: The revised sentence reads (L381ff): “Our findings suggest, but do not prove, that web-building spiders might do this as well. With the pheromone system of *S. grossa* now known, potential manipulation by female spiders of their webs’ pH, and thus their webs’ attractiveness to mate-seeking males, can now be tested in the context of honest or dishonest signalling”. Please see also R4.

15. 355-369 I found this paragraph somewhat speculative, not needed in such a high level and well supported manuscript. If the chemicals that trigger courtship transition the chemicals that attract mates, does it mean that older webs will attract mates but will not trigger courtship? How long does a web of *Steatoda grossa* last in nature? Please address the issue.

R15: As suggested, we have deleted this entire (admittedly speculative) paragraph. We will address the question as to how long a web of *Steatoda grossa* lasts in nature and remains attractive in another manuscript that is currently in preparation.

16. 461 - “the male engaged in courtship, pulling silk with his hindlegs from his spinnerets and adding it to the paper.” This sentence deserves a citation and probably a very brief explanation. Pulling silk as a proxy of courtship will probably sound weird, since only those working on the taxon and subject know the sexual behavior of theridiids and others of silk wrapping. Please cite the works of Catherine Scott and/or Andreas Fischer on courtship in *Steatoda grossa*.

R16: Good point. We have added an explanatory sentence (L491ff): “Sensing contact pheromone, the male essentially behaves as if he were courting on the web of a female. On a web, the male engages in web reduction prior to copulation, a behaviour that entails cutting sections of the female’s web with his chelicerae and wrapping the dismantled web bundle with his own silk pulled from his spinnerets”. We have cited the work by Dr. Catherine Scott and Andreas Fischer.

17. 514-521 – It seems that the authors have used the same spiders in day 1 and day 14. “After female spiders had spun their webs on bamboo scaffolds for three days, they were removed and webs were extracted immediately (0-day-old webs) or following 14 days of aging (14-day-old webs).” If that is the case, please specify that you have used Wilcoxon signed rank (matched pair design) and not Wilcoxon rank sum (independent samples).

R17: We used independent webs. Each spider produced a single web in this experiment. Thus, the Wilcoxon rank sum test (for independent samples) is appropriate for data analyses. For clarification, we have revised the section to read (L548ff): “To test for the hydrolysis of the contact pheromone components **12**, **16** and **17**, we compared their breakdown ratio ($\mathbf{18 / (12 + 16 + 17 + 18)}$) on independent webs aged 0 days and 14 days (Exp. 24). Each of 140 spiders was allowed to spin a web on bamboo scaffolds for three days. Then, the spiders were removed and webs – by random assignment – were extracted immediately (0-day-old webs) or after 14 days of aging (14-day-old webs).

18. 537 – Spiders leave a dragline, often with pheromones, when they walk. In the Y olfactometer experiment, the authors did not mention cleaning the apparatus after each trial. They also did not mention how they scored the trials. What is the operational variable for “being attracted” (L 234)? Each researcher uses a distinct criterium. Often, arthropods go to the end of one of the arms of the Y maze, go back, move to the end of the other arm and remains there, for example.

R18: We have added the formerly missing information (L569ff): “A male that entered the olfactometer within the 5-min bioassay period was classed a responder and his first choice of oven bag (the oven bag he reached first) was recorded; “Whenever a set of replicates was completed, the Y-tubes were cleaned with hot water and soap (Sparkleen, Thermo Fisher Scientific, MA, United States) and dried in an oven at 100 °C for three hours, whereas the bamboo sticks and the oven bags were discarded. “We have further expanded the information on test stimuli (L560ff): “Test stimuli consisted of a triangular bamboo prism scaffold (each side 8.5 cm long) bearing a spider’s web, or bearing artificial webbing³⁰ (40 ± 2 mg; Bling Star, CN) that was treated with web extract or synthetic chemicals in methanol (100 μ L) as the treatment stimulus or with methanol (100 μ L) as control stimulus.”

19. 637 – The authors have used, in a single case, a one tailed test. They have justified it with the citation of a reference. Please elaborate on it. Other tests ran in the study had a “one-tailed” context, expectation, a hypothesis that generate a prediction of one group having smaller or larger values than the other. But in such cases, two tailed tests were used.

R19: We analyzed all data collected in Y-tube olfactometer experiments 22, 23, and 25-27, where we predicted attraction of spiders to pheromone treatment stimuli but not to solvent control stimuli, with a one-sided test. We contend that Reviewer #1 agrees with this choice of statistical test. We deemed one-sided tests less appropriate for all the other experiments. For example, in T-rod experiments, we tested the responses of spiders to two treatment stimuli, rather than to one treatment stimulus and one control stimulus, making it difficult to predict the outcome of experiments. Thus, we opted to analyse those data using a more conservative two-tailed test. We have provided explanations in the section ‘Statistical analyses of data’, as follows (L675ff): “Data of experiments 19–21 (revealing the presence of contact pheromone components in the abdomen, silk glands, and posterior aggregate silk gland) were analysed with two-tailed, rather than one-tailed, Wilcoxon test or Kruskal-Wallis rank tests because we had no strong assumption as to whether or not pheromone would be present in any of these potential pheromone sources; the p-values were adjusted for multiple comparison using the Benjamini-Hochberg method. Y-tube olfactometer data of experiments 22, 23, 25–27 as well as data of hallway experiment 28 (revealing attraction of male spiders to volatile pheromone components) were analysed using an one-tailed⁷⁵ binomial test, anticipating attraction of spiders to volatile mate attractant pheromone components rather than to solvent control stimuli”.

Reviewer #2 (Remarks to the Author):

20. Fischer et al. identified the chemicals of contact sex pheromones of a widow spider species and demonstrated their roles in attracting males. This study also provides evidence showing that the breakdown of a contact pheromone leads to chemicals with higher volatility which can attract males over a distance. This is a very cool study and the manuscript is well written with clear visualization. I only have some minor comments for this manuscript and I recommend a minor revision.

R20: We thank Reviewer 2 for the positive assessment of our manuscript.

Minor comments.

21. • Technically speaking, spiders do not belong to insects. While in the second paragraph of the Introduction, the authors mainly introduce pheromone-related literature in insects but ignored findings in a broader range of species in arthropods. I suggest the authors have a more inclusive introduction on arthropods, instead of only focusing on insects.

R21: Good point. We have added a range of animal taxa that are known to communicate via pheromones. The revised section reads (L30f): “Chemicals such as pheromones are deemed the oldest form of (sexual) communication signals⁸ and have evolved in various animal taxa including mammals¹⁶, myriapods¹⁷, crustaceans¹⁸, and insects¹⁹⁻²².”

22. • Line 53-54, this study only showed that pH could lead to different efficiencies in the breakdown of contact pheromone but not how female spiders actively modulate this breakdown. I suggest rephrasing this sentence to make it clearer.

R22: In lines 53-54, we have simply listed what is not known. We are not claiming that we have definitive answers for all of these unexplored questions. If agreeable, we prefer not to revise the text.

23. • Line 55, this study did not investigate how mate-seeking males sense pheromones. I suggest the authors delete this sentence.

R23: As suggested, we have deleted the sentence.

24. • Figure 5, how about putting panel b to the left side of panel a?

R24: We have carefully considered this suggestion but prefer the original (vertically stacked) arrangement that allows alignments of the x-axes in subpanels **a** and **b**.

Reviewer #3 (Remarks to the Author):

25. This is a very nice paper on the pheromone communication of the false black widow spider. The authors identified a non-volatile sex pheromone produced by females and applied onto the web to induce courtship in males. The exiting and new part of the study is, that the same non-volatile compounds are catalyzed by an enzyme into a volatile form to attract males over longer distances. Thus, the paper adds not only a lot of new information on the pheromone communication of spiders but reveals a new mechanism that connect the two pheromones and produces the volatile form. Also, from an evolutionary point of view, this is very interesting. Technically, the paper is well done. The authors put a lot of work in the chemical analysis and especially synthesis of the pheromone compounds. Also, the behavioral experiments are quite extensive. I only have a few minor comments, mostly on figures and statistical analysis:

R25: We thank Reviewer 3 for the complimentary comments.

26. Fig. 1e: If the statistic is ranked based, i.e. a Kruskal-Wallis or Wilcoxon test, I think it is not really appropriate to show mean and SE, but better to give a box plot (combined with datapoints). This also applies Fig. 2d, 4d and 5b.

R26: Revised as suggested.

27. Fig. 2d: You obviously also use pairwise Wilcoxon Test after the Kruskal-Wallis test. Did you also use a correction for multiple comparisons? Also, the background picture is nice, but does not add any information. Please remove.

R27: Yes, we have applied the Benjamini-Hochberg correction to adjust for multiple comparisons. We have added the info to the revised text (L103): “Medians with different letters indicate statistically significant differences in courtship responses; Kruskal-Wallis χ^2 test with Benjamini-Hochberg correction to adjust for multiple comparisons, $P < 0.05$ ”. The picture displays web bundling behaviour by a male and visually relates to the data that are presented. For that reason, we would like to retain the picture but we defer to Dr. Gene Chong as the Editor to make the final decision.

28. Line 196 and Fig. 3b. The 140 samples have been taken from 20 spiders and are thus not independent. Better write $N=20$.

R28: Good point. Revised accordingly.

29. Line 198: Same as above. $N=30$.

R29: Revised accordingly.

30. Line 210 and Fig. 3: Did you use Fisher's exact test for a), b) and c)? Why this test and not a Kruskal-Wallis? If you used Fisher's exact test, you did only test the presence or absence of the compounds, right? This should be stated. And does not really fit with a figure showing the amounts and means.

R30: Good point, again. In the revised manuscript, we have now analysed the quantitative data with non-parametric tests. We have revised the pertinent text to read (L675ff): “Data of experiments 19–21 (revealing the presence of contact pheromone components in the abdomen, silk glands, and posterior aggregate silk gland) were analysed with two-tailed, rather than one-tailed, Wilcoxon test or Kruskal-Wallis rank tests because we had no strong assumption as to whether or not pheromone would be present in any of these potential pheromone sources. The p-values were adjusted for multiple comparison using the Benjamini-Hochberg method”.

31. Fig. 4f: You put up 10 traps (line 555), right? So N=10, not 160. Furthermore, what is shown in the figure? The mean per trap or the total number of spiders caught in all traps? Which statistics did you use?

R31: Yes, at any time 10 trap pairs were concurrently run which were replaced every week between September and December. To avoid ambiguity, we have revised the text accordingly (L276f): “(f) Captures of *S. grossa* males in 10 pairs of sticky traps that were deployed in building hallways between September and December 1018. During weekly checks, the position of the treatment and control trap within each pair was randomized; the treatment trap was baited with the carboxylic acids **19**, **20**, and **21** (see Methods for detail), whereas the control trap was left unbaited; the asterisk denotes a significant preference for the treatment trap (one-tailed binomial test; $P < 0.05$)”. We have revised the label of the x-axis to “Total number of males captured”. In the Method section, we have expanded the info (L601f): “Every week for four months (September to December 2018), traps were checked, lures replaced, and the position of the treatment and the control trap within each trap pair was re-randomized”.

32. Line 261: Move the reference for table S1 right behind the compound numbers.

R32: Right behind compound names, we have added ‘(see methods for detail)’ because the methods describe how chemicals were formulated.

REVIEWERS' COMMENTS:

Reviewer #1 (Remarks to the Author):

The authors have addressed all my comments adequately but I would ask one last thing. They have now clarified that they ran experiments from 9-17h (8h per day). For the 4 Y maze experiments, they used a total of 108 animals. They have now explained that there was a 3h interval between trials due to cleaning of the Y mazes. Time required is therefore 324h, which is equivalent to ~40 days considering 8 full hours per day. Excluding weekends, it means ~2 months only of intervals between trials. If all these trials were run in the summer 2018, I suppose the authors used several Y mazes and had several researchers collecting Y mazes data. I think it would be important to mention the number of Y mazes used and researchers involved. Thank you.

Reviewer #3 (Remarks to the Author):

In the revised manuscript, the authors carefully addressed all question and remarks from my review. I have not further comments and recommend publication.

Responses (R) to reviewers' Comments

1. Reviewer #1 (Remarks to the Author):

The authors have addressed all my comments adequately but I would ask one last thing. They have now clarified that they ran experiments from 9-17h (8h per day). For the 4 Y maze experiments, they used a total of 108 animals. They have now explained that there was a 3h interval between trials due to cleaning of the Y mazes. Time required is therefore 324h, which is equivalent to ~40 days considering 8 full hours per day. Excluding weekends, it means ~2 months only of intervals between trials. If all these trials were run in the summer 2018, I suppose the authors used several Y mazes and had several researchers collecting Y mazes data. I think it would be important to mention the number of Y mazes used and researchers involved. Thank you.

R1: Thank you for having pointed this out. You are correct assuming that we had multiple Y-tubes for bioassays, thus facilitating time- and work-efficient bioassay testing and cleaning procedures. We have revised the text accordingly (lines 547-550): "Whenever a set of 30 replicates was completed by the same observer, using 30 separate Y-tubes, the Y-tubes were cleaned with hot water and soap (Sparkleen, Thermo Fisher Scientific, MA, United States) and dried in an oven at 100 °C for three hours, whereas the bamboo sticks and the oven bags were discarded".

2. Reviewer #3 (Remarks to the Author):

In the revised manuscript, the authors carefully addressed all question and remarks from my review. I have not further comments and recommend publication.

R2: Thank you, again, for the constructive comments. They have helped improve the manuscript.